behaviour

cooperation, enforcement, punishment, pay-to-stay, negotiations, cooperative breeding

**Author for correspondence:**
Jan Naef
e-mail: janusnaefus@gmx.ch

# Commodity-specific punishment for experimentally induced defection in cooperatively breeding fish

## Jan Naef and Michael Taborsky

Department for Behavioural Ecology, University of Bern, Wohlenstrasse 50a, Hinterkappelen CH-3032, Switzerland

JN, 0000-0001-6319-8753

Coercion is an important but underrated component in the evolution of cooperative behaviour. According to the pay-to-stay hypothesis of cooperative breeding, subordinates trade alloparental care for the concession to stay in the group. Punishment of idle subordinates is a key prediction of this hypothesis, which has received some experimental scrutiny. However, previous studies neither allowed separating between punishment and effects of disruption of social dynamics, nor did they differentiate between different helping behaviours that may reflect either mutualistic or reciprocal interaction dynamics. In the cooperative breeder *Neolamprologus pulcher*, we experimentally engineered the ability of subordinates to contribute to alloparental care by manipulating two different helping behaviours independently from one another in a full factorial design. We recorded the treatment effects on breeder aggression, subordinate helping efforts and submissive displays. We found two divergent regulatory mechanisms of cooperation, dependent on behavioural function. Experimental impediment of territory maintenance of subordinates triggered punishment by dominants, whereas prevented defence against egg predators released a compensatory response of subordinates without any enforcement, suggesting pre-emptive appeasement. These effects occurred independently of one another. Apparently, in the complex negotiation process among members of cooperative groups, behaviours fulfilling different functions may be regulated by divergent interaction mechanisms.

# 1. Introduction

Cooperation theory deals with the central paradox that interactions may involve altruism, which ostensibly cannot evolve by natural

selection [1]. Four solutions have been proposed to resolve this paradox [2]. (i) Kin-selection theory [3,4] explains cases where cooperating partners have common fitness interests due to shared genes, as exemplified by eusocial organisms. (ii) In mutualistic interactions such as in group hunting predators, individuals may benefit each other as a by-product of purely selfish traits [5,6]. (iii) Alternatively, cooperative behaviour may be demanded from individuals through enforcement by more powerful social partners [5,7,8]. (iv) Finally, evolutionarily stable levels of cooperation can result from reciprocal exchanges among cooperating partners [9,10]. These four evolutionary mechanisms are not mutually exclusive. Especially, the exchange of different commodities and services in asymmetrical relationships may involve both reciprocity and coercion [11,12]. The dominant partner can enforce cooperative behaviour in the subordinate through threats of aggression [8,13]. The subordinate, on the other hand, may have outside options and threaten to end the cooperative interaction altogether, thereby putting a limit on what dominants can demand without reciprocating [13,14]. In social cichlids and paper wasps, for example, subordinates provide less help if they are given options to switch to a different group [15–17].

The interplay of coercion and reciprocity is of particular importance in complex societies where individuals of different relatedness and dominance exchange commodities in long-term social relationships, as has been demonstrated in chimpanzees, bats and other species serving as models for the study of reciprocal cooperation [11,18,19]. In many such societies, affiliative behaviour and ritualized aggression–submission interactions may have evolved to minimize conflict by establishing a clear dominance hierarchy [20]. Dominance interactions and reciprocal exchanges can be confounded in three ways. First, the power asymmetry inherent in these dominance relationships can influence reciprocal exchanges between two partners by affecting their relative negotiating power [12]. Second, reciprocal exchanges between two partners can in turn influence their dominance relationship. In vervet monkeys, for example, the dominance rank of subordinates increased after they were given the opportunity to provide food for their group [21]. Third, affiliative behaviour and aggression–submission interactions can be observed both in the context of dominance interactions and reciprocal exchanges [22]. Drawing a clear line between dominance interactions and reciprocal exchanges can therefore be difficult and, in fact, misleading [22–24].

Cooperative breeders offer a unique opportunity to study reciprocal exchanges in asymmetrical relationships. They form groups in which dominant individuals monopolize reproduction, while subordinates help to raise the dominant's offspring [25–28]. Often, subordinates are not related to the offspring they help to raise [29–32]. In this case, the costs of helping [33,34] need to be balanced by direct fitness benefits, which may include trading of commodities between group members [11]. Acceptance in a group, for instance, may depend on fulfilling demands of dominant group members, which has been modelled by the pay-to-stay hypothesis of cooperative breeding [35–37]. This hypothesis proposes that subordinates pay with alloparental care for the concession to stay in the territory of dominant breeders, which implies that breeders accept or evict helpers depending on their payment (see [38] for a review). Subordinates may benefit in many ways from being tolerated by territory owners, for instance by resource access, reproductive participation and through reduced predation risk [39–44].

The pay-to-stay hypothesis is based on the assumption that helping has evolved through a function similar to submissive behaviour: it inhibits aggression in dominant group members [40,45,46]. In accordance with this hypothesis, unrelated helpers provide more help than related ones [47,48], and experimental reduction of helping may lead to a subsequent increase in helping, submission and punishment [45,49–52]. Previous studies supporting the pay-to-stay hypothesis suffer from two important shortcomings. First, they have not adequately controlled for disruptions of social dynamics. In some cases, helping behaviour has been manipulated by removing a helper from the group [49,50] or by confining it to a small or remote part of the territory [45,52]. Without adequate controls, such experiments cannot distinguish between effects of reduced helping and effects of disrupted social interactions [53,54]. Second, they have not differentiated between the many helping behaviours that subordinates may perform. Brood care, defence against a wide range of challengers, and territory maintenance by subordinates are all categorized as helping because they are beneficial for dominants and costly to subordinates [38,55,56]. This may lead to a confusion of mutualistic interactions with reciprocal exchanges, because some of the behaviours classified as helping may simply be maintained through immediate direct benefits to the subordinate. For example, maintaining the breeding shelter may be beneficial for subordinates because they gain space for evading predators, but it simultaneously decreases the mortality of unrelated young [57]. Thus, some of the behaviours classified as helping may reflect a by-product mutualism [5], whereas others rely on a give-and-take basis reflecting reciprocity [11]. Experiments that manipulate several helping behaviours at the same time cannot distinguish between these two hypotheses.

Here, we investigate the social regulation of two putative helping behaviours in a cooperatively breeding cichlid fish, paying special attention to the distinction between mutualistic interactions and reciprocal exchanges, and the interplay between coercion and reciprocity. We aim to unveil which specific helping behaviours are causally involved in the regulation of aggressive behaviours of dominants and are thus crucial elements of the reciprocal exchange predicted by the pay-to-stay hypothesis. We manipulated the two helping behaviours independently from one another and without impeding social interactions, and we measured how these manipulations affected (i) the aggression of dominants towards the experimental subjects, (ii) the submissive behaviour of the manipulated subordinates, and (iii) their potential compensation in response to the temporary helping inhibition. Specifically, while allowing the focal helper to interact with other group members and to access the centre of the territory, we prevented them from performing two behaviours proposed to function as rent payment: digging sand out of the breeding shelter, and defending the territory against an intruding egg predator.

# 2. Methods

## 2.1. Study species

*Neolamprologus pulcher* is a small plankton-feeding cichlid fish endemic to Lake Tanganyika, Africa [58,59]. These cooperative breeders live in groups consisting of a breeding pair and up to 30 helpers with a strict size-based dominance hierarchy [39,57,60,61]. Groups defend a number of shelters dug out from sand under rocks, and aggregate in colonies of up to 200 groups [62,63]. Relatedness among group members declines with their age [29], and reproduction is largely monopolized by the breeding pair [31,38].

*Telmatochromis vittatus* is an opportunistic predator of eggs and fry [64–67] occurring at high abundance within *N. pulcher* colonies [62]. *Neolamprologus pulcher* helpers defend the territory against *T. vittatus* even though they are no direct threat to them, which is why this defence has been determined as altruistic helping behaviour primarily benefitting the dominant breeders [66–68].

## 2.2. Experimental animals

We used 96 *N. pulcher* as focal subjects and 20 *T. vittatus* as egg predators from our laboratory stock population originating from Kasakalawe point, Zambia. *Neolamprologus pulcher* stock was kept in 400 l tanks as separate-sex aggregations of about 30 individuals without breeding shelters. This simulates naturally occurring aggregations of non-breeding individuals growing fast before getting large enough to take over a breeding position [39,57]. Egg predators were kept in groups of about 30 in 200 l tanks with breeding shelters. All aquaria were maintained at 26°C, with a 13 : 11 h light : dark cycle. Fish were checked daily and fed with dry food 5 days a week, and with frozen plankton on 1 day. All aquaria contained air-driven biological filters.

## 2.3. Breeding groups

We created 32 groups of one large male (5.5–6.5 cm SL) and two females (4.5–5.5 cm SL, 3.5–4.5 cm SL), the larger of which would become the breeding partner of the male and the smaller one their helper. Groups with only one helper are relatively rare in the wild [60], but when created in the laboratory they show natural behaviour. In order to achieve a clear dominance hierarchy, helpers were chosen to be at least 10 mm smaller than the dominant females and the latter at least 10 mm smaller than the dominant males. All fish were caught from aggregation tanks, measured, weighed, sexed and assigned to groups according to their size. Helpers were put in the territory first (day 0) and took possession of the presented shelter within minutes. The dominant female and the male were introduced in small isolation nets on the same day as the helper and released on the next day (day 1) and the day after next (day 2), respectively. The fish usually engaged in aggressive interactions initially upon release, but established a clear dominance hierarchy until day 3, when ritualized aggression–submission interactions predominated and the helper was accepted in the breeding shelter. As digging behaviour is initially infrequent in fish taken from an aggregation, the shelters of all groups were filled with sand to two-thirds of the height prior to the start of the experiment, once each on days 3–6, which stabilized digging levels in all groups. From among the groups that remained stable on day 6, 24 were randomly selected for the experiment. Manipulations and observations took place on days 9–12.

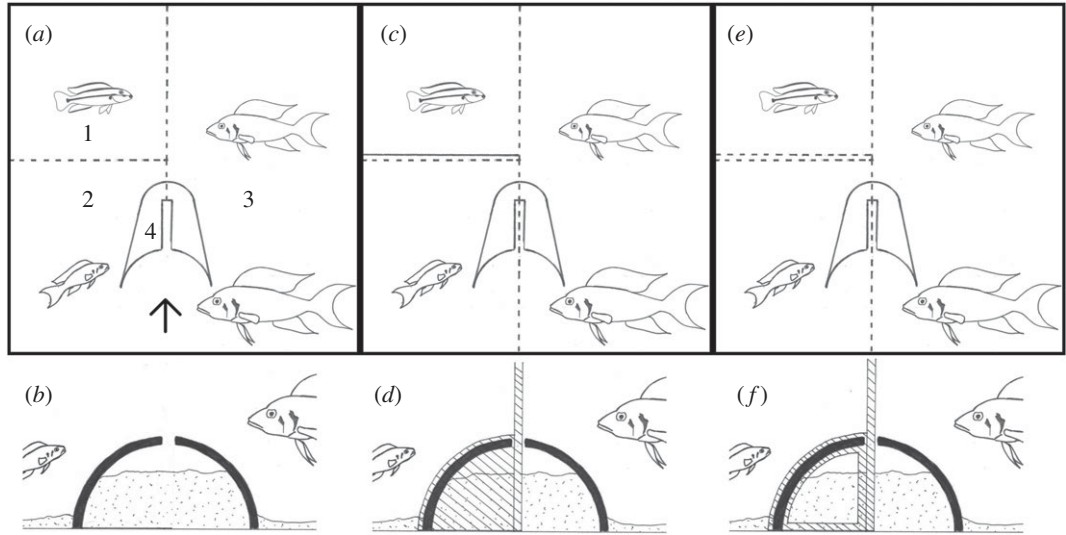

**Figure 1.** Manipulation of defence (top row) and digging behaviour (bottom row). (*a*) The basic tank set-up seen from above, without experimental manipulations. Dotted lines depict transparent partitions with holes. 1, Intruder presentation compartment; 2, Helper compartment; 3, Breeder compartment; 4, Breeding shelter. (*b*) Detail of (*a*) as seen from the direction of the arrow, showing the breeding shelter (solid black) filled with sand (dotted). The gap in the top of the shelter accommodates a transparent partition to separate the helper from the breeders during manipulations. This partition is shown in (*c*) and (*e*) as a dotted line, and in (*d*) and (*f*) as a hatched line passing through the gap in the shelter. (*c*) Defence behaviour is prevented by inserting an opaque partition (solid line) between helper and intruder. (*e*) To allow defence and control for manipulations, a transparent partition (dotted line) is used instead. (*d*) Digging behaviour is prevented by blocking the entrance to the shelter in the helper compartment with a precisely cut transparent partition (hatched). (*f*) To allow digging and control for manipulations, a partition with a large hole is used instead.

## 2.4. Aquarium set-up

Each breeding group was established in one half of a 200 l tank, separated from the group in the other half by a clear partition with holes, such that each group could defend their territory against another group. This greatly reduces within-group aggression and increases group stability, compared to keeping each group separately. During experiments, we inserted an opaque partition between the focal group and the group in the other half of the tank to prevent interactions with members of the neighbouring group. Each group's territory was divided into three compartments, one for presenting an egg predator, one for the helper, and one for the breeders (figure 1*a*). A flowerpot half representing the breeding substrate to which our laboratory fish are used was cut and placed in such a way that half of the breeding chamber was in the helper's compartment and the other half in the breeders' compartment (figure 1*a,b*). Outside of experimental periods, there was no partition between helper and breeder compartments, such that all fish could freely use both of these compartments and the entire shelter.

## 2.5. Treatments

During each treatment, the breeding shelter was filled with sand to two-thirds of the height (figure 1*b*) to induce digging, and one *T. vittatus* was presented in the intruder compartment to induce territory defence. The helper's ability to contribute to territory maintenance (digging sand out of the shelter) and defence (attacking the intruder) was then manipulated. In order to manipulate the helper's ability to dig or to defend, we separated it from the breeders with help of a clear partition. We prevented digging behaviour by closing the shelter entrances in the helper compartment, using a precision-cut transparent acrylic plate (figure 1*d*). Where digging was allowed, this plate had a large hole through which the helpers entered the shelter without hesitation and continued to dig as usual (figure 1*f*). Defence behaviour was prevented by obstructing the view from the helper compartment to the intruder compartment with an opaque partition (figure 1*c*). Where defence was allowed, a transparent partition was inserted instead (figure 1*e*). Both of these partitions had holes that lined up with the holes in the fixed partition between intruder and helper compartments, to allow for passive water exchange. Since these cichlids rely heavily on chemical communication [69–71], we assume that the helpers have recognized by smell when an intruder was

present. Antipredator aggression and social interactions including aggressive, affiliative and submissive behaviours are readily exchanged between group members through clear partitions. Territory maintenance (digging out the shelter) and defence (attacking the egg predator) manipulations were combined to yield four distinct treatments: N (Nothing prevented; helper can participate in both territory maintenance and defence), D (defence behaviour is prevented), M (maintenance behaviour is prevented) and DM (both are prevented). Each group experienced all four treatments, one each on days 9–12, at a fixed hour and with the same intruder. The 24 possible treatment orders were assigned randomly to the groups, each treatment order being used once.

After all treatments were recorded, we performed an additional control experiment on half of the groups. In this 'no demand' control, the groups were exposed to the same procedure as during the original experiment, except that no intruder was presented, and the shelter was not filled with sand. Each group received all four treatments, one each on days 13–16, in random order. This control served to test whether the manipulations of helping behaviour would influence social interactions in any unforeseen way, for example by restricting the view or movement of the helper, or the interactions between group members.

## 2.6. Experimental procedure

For each treatment, we first filled the breeding shelter with sand to two-thirds of the height and simultaneously introduced the intruder in the intruder compartment. During the following 10 min, we left the fish undisturbed and concealed the intruder behind a black partition to accommodate it to the new environment. Then we confined the helper in its compartment, removed the partitions concealing the intruder, and installed the treatments as described above. With the treatments installed, the group was left undisturbed for 12 min while we recorded it with two video cameras ('manipulation phase', cf. Figure 1c–f). After that, we removed all partitions that manipulated the helping behaviour, thereby releasing the helper from its confinement and re-establishing the situation the group was used to, except for the sand in the shelter and the presence of the intruder. In this situation, where the helper could swim freely and perform defence and digging behaviour, the group was again left undisturbed and recorded for 12 min ('test phase', cf. figure 1a,b). Finally, we removed the intruder and commenced working with the next group.

## 2.7. Behavioural observations

We used two SJCAM M10 cameras to record videos of the fish at 720p, 30 fps and 6554 kb s$^{-1}$, compressed as mpeg-2. One camera was placed in front of the aquarium, and the other one on top of the helper compartment. The two video streams were synchronized using a clapperboard and stored in a single matroska container file using the open-source software tools FFmpeg [72] and Audacity® [73]. The two 12 min parts needed for the analysis were then extracted, taking care to cut out any information about group identity or treatment from the start or end of the videos, and stored in a separate folder with uninformative filenames. The observer would thus be blind to the independent variables of the treatment during scoring of the test phase. During the manipulation phase, the installed experimental partitions were inevitably visible. Behaviours were scored from the videos using the free software BORIS [74]. We recorded all behaviours of the helper and all helper-directed behaviours of both breeders. The ethogram included aggressive displays (operculum spread, fin spread, lateral display, head down display), overt aggression (biting, ramming), submissive displays (tail quivering and backwards approach), affiliative behaviour (bumping) and territory maintenance (digging) (cf. [39]). Displays were recorded as events with duration, all other behaviours as point events. All behaviours were eventually analysed as counts. Counts of overt aggression and aggressive displays by the helper towards the intruder were summed up as defence behaviour, those of both breeders towards the helper as breeder aggression.

## 2.8. Data analysis

We used raw frequencies of behaviours (aggressive behaviours of breeders to helpers, submissive displays of helpers, digging behaviours of helpers and defence behaviours of helpers) per time (the entire 12 min recording period, and in some cases, the initial 6 min; see Results) as dependent variables, the two manipulations (manipulation of territory maintenance and manipulation of antipredator defence) as independent variables and group ID as blocking variable. All dependent variables followed a negative binomial distribution. The interaction between treatments was dropped from all models because it was not significant. The four datasets obtained from this experiment

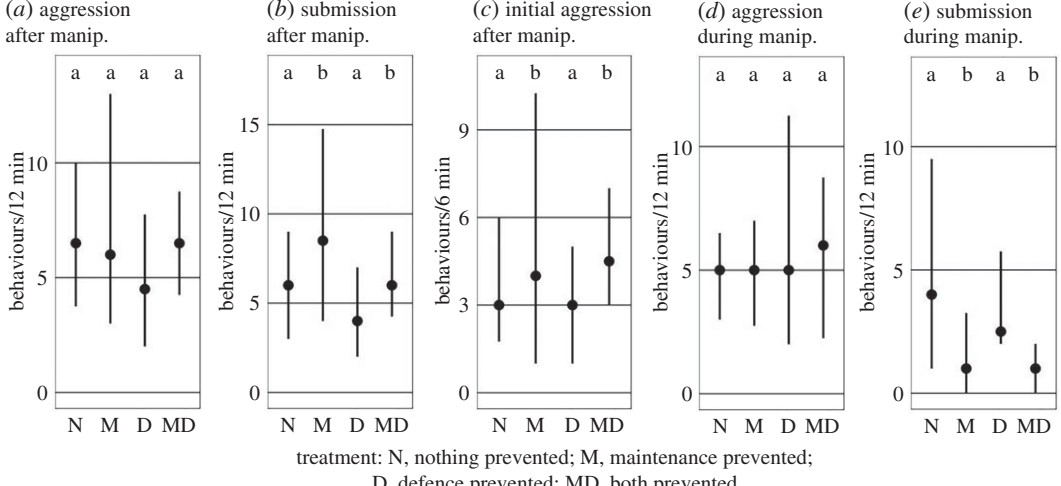

**Figure 2.** Effects of manipulating helping behaviours on social interactions (aggressive behaviours of breeders to helpers and submissive displays of helpers) during the 'test phase' (*a*–*c*) and during the 'manipulation phase' (*d,e*). Means and interquartile ranges are shown.

(manipulation phase and test phase of experimental trials and 'no demand' control trials, respectively) were analysed separately in a two-step procedure. We first conducted a non-parametric multivariate test on each dataset to assess treatment effects on behavioural composition (PERMANOVA from the R-package vegan [75], using Gower dissimilarities [76]). Where significant effects were detected, we performed follow-up analyses of individual behaviours with generalized linear mixed models for negative binomial data using the R-package lme4 [77]. All data processing was done using RStudio [78].

# 3. Results

## 3.1. Treatment effects on behavioural composition

In the experimental trials, manipulating defence and digging behaviour of the helper significantly influenced the behavioural composition in the 'test phase' (preventing maintenance: d.f. = 1, $F = 2.88$, $p = 0.006$; preventing defence: d.f. = 1, $F = 2.31$, $p = 0.023$). Preventing maintenance influenced behavioural composition also during the 'manipulation phase' (d.f. = 1, $F = 4.99$, $p = 0.001$), but preventing defence did not (d.f. = 1, $F = 0.42$, $p = 0.622$).

## 3.2. Treatment effects on social interactions

Preventing the digging behaviour of subordinates caused a 30% increase of dominants' aggression (figure 2*a*, est. = 1.28, $p = 0.0576$) and a 50% increase in subordinate submission (figure 2*b*, est. = 1.56, $p = 0.0003$) during the subsequent 'test phase'. Of the 640 aggressive behaviours observed during the entire 12 min 'test phase', 424 (66.3%) occurred in the first 6 min. During those initial 6 min, preventing digging behaviour caused a 50% increase in the dominants' aggression ('initial aggression'; figure 2*c*; est. = 1.42, $p = 0.023$). During the 'manipulation phase', preventing digging did not influence breeder aggression (figure 2*d*, est. = 1.06, $p = 0.736$), but subordinates showed only half as much submission if they were prevented from digging (figure 2*e*, est. = 0.40, $p = 0.00002$).

 Preventing the defence behaviour of subordinates had no significant influence on breeder aggression or subordinate submission in the subsequent 'test phase' (figure 2*a*–*c*; aggression: est. = 0.84, $p = 0.176$; submission: est. = 0.80, $p = 0.071$; initial aggression (aggression during the first 6 min of the test phase (see above): est. = 0.94, $p = 0.706$).

## 3.3. Treatment effects on helping behaviours

Preventing the digging behaviour of subordinates had no effect on digging during the subsequent 'test phase' (figure 3*b*, est. = 1.14, $p = 0.729$), and it affected defence behaviour neither during the 'test phase'

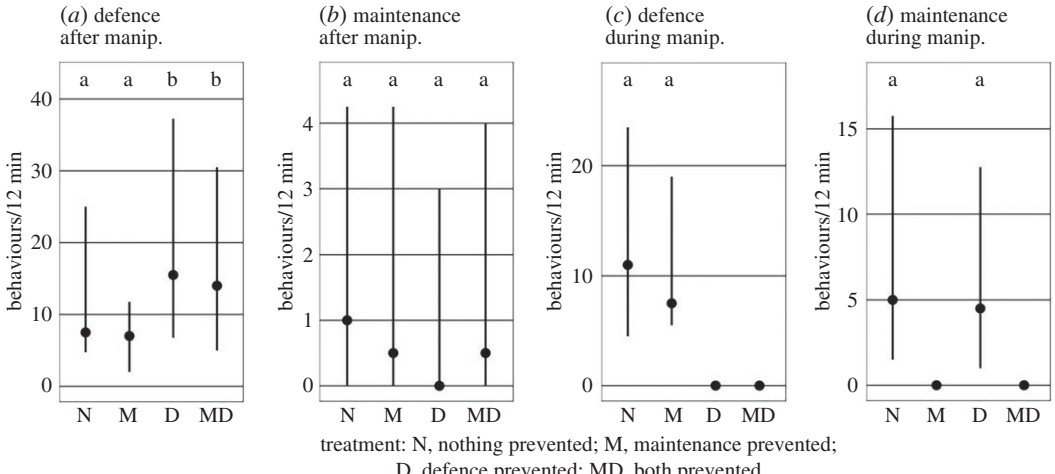

**Figure 3.** Effects of manipulating helping behaviours on helping behaviours (defending the territory against an egg predator and maintaining the breeding shelter by digging out sand) during the 'test phase' (*a,b*) and during the 'manipulation phase' (*c,d*). Means and interquartile ranges are shown. Note that the frequency of defence behaviours in (*c*) is zero in two treatments because defence was experimentally prevented. The same is true for maintenance behaviour (digging) in (*d*).

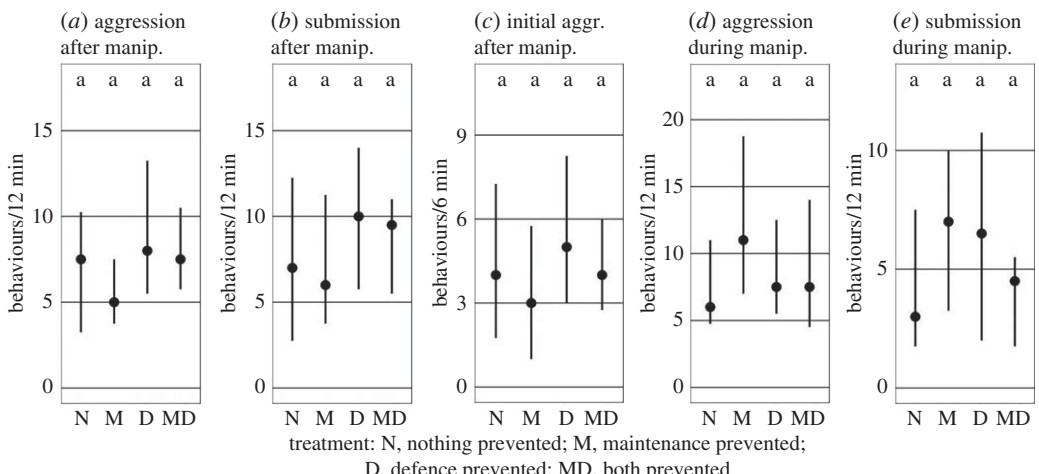

**Figure 4.** Effects of the manipulating procedure on social interactions (aggressive behaviours of breeders to helpers and submissive displays of helpers) during the 'test phase' (*a–c*) and during the 'manipulation phase' (*d,e*) in a control experiment without demand for help ('no demand' control). Means and interquartile ranges are shown.

(figure 3*a*, est. = 0.85, $p = 0.198$) nor during the 'manipulation phase' (figure 3*c*, est. = 1.01, $p = 0.96$). Preventing the defence behaviour of subordinates caused a 50% increase in their defence behaviour during the subsequent 'test phase' (figure 3*a*, est. = 1.65, $p < 0.00001$). Defence efforts of subordinates when they first saw the predator were also higher after they had been prevented from defending (comparison of D and DM treatments, 'test phase', to N and M treatments, 'manipulation phase': est. = 1.37, $p = 0.038$). By contrast, preventing the defence behaviour of subordinates did not affect their digging behaviour during the 'test phase' (figure 3*b*, est. = 0.67, $p = 0.307$).

## 3.4. Control for potential effects of the manipulation procedure

In the 'no demand' control trials, neither manipulation affected behavioural composition. This pertains for both the 'test phase' (figure 4*a,b*; preventing maintenance: d.f. = 1, $F = 0.32$, $p = 0.685$; preventing defence: d.f. = 1, $F = 0.85$, $p = 0.344$) and the 'manipulation phase' (figure 4*d,e*; preventing maintenance: d.f. = 1, $F = 0.34$, $p = 0.53$; preventing defence: d.f. = 1, $F = 0.24$, $p = 0.621$). Since in the experimental trials, breeder aggression during the test phase was concentrated in the first half of the measuring period and analysed separately ('initial aggression', see above), we tested for a similar effect in the

control treatments. Neither manipulation affected initial aggression in the control treatments (figure 4c; preventing digging behaviour: est. = 0.87, p = 0.449; preventing defence behaviour: est. = 0.98, p = 0.906).

## 4. Discussion

Our data show that dominant breeders adjust their level of aggression towards subordinates depending on how much the latter invest into territory maintenance (i.e. costly digging behaviour [56]). Dominants increased aggression by roughly 50% after subordinates were prevented from digging, and subordinates showed a corresponding increase in submissive displays. By contrast, the dominants' aggression and subordinate's submission were not affected by an experimental prevention of territory defence by subordinates, neither during nor after manipulations. However, subordinates increased their defence effort after defence had been prevented. We found no evidence that territory maintenance and defence are co-regulated. Preventing digging had no influence on defence behaviour and vice versa, neither in the manipulation nor in the test phase. In accordance with this, the effects that our manipulation of digging behaviour had on breeder aggression and subordinate submission were independent of our manipulation of defence behaviour.

Punishment of idle helpers has been the focus of several previous studies of cooperative breeders [49,50,52], but so far it has been unclear whether the increased aggression observed in response to the experimental manipulation resulted from enhanced conflict over rank or territory, as the manipulation involved temporary removal or confinement of the helper [49,50,52,54]. The present study also used partial confinement of subordinates to manipulate their helping behaviour, since particular behaviours cannot be manipulated otherwise. However, we improved over previous experiments in three ways: First, helping behaviours were manipulated independently from the confinement situation of the helper. All treatments included the same confinement procedure, regardless of whether helping was possible or prevented. Second, the potential effects of confinement on social behaviour were minimized. During confinement, the focal subordinate could swim freely in approximately half of the territory including the central breeding shelter, and interact with other group members through a transparent partition, which happened frequently. Note that this is not necessary for distinguishing between the effects of confinement and manipulations of helping behaviour, but intended to maintain normal social behaviour during the manipulation phase. Third, in the 'no demand' control experiment, we excluded the possibility that our manipulations of helping behaviour affected social behaviour in ways unrelated to reduced helping. For example, our manipulation of digging behaviour prevented not only digging, but all shelter visits of the subordinate, which might also affect its other behaviour. This control used the same manipulations as the original experiment in a situation where no help was required. The manipulations had no effect on social interactions in this control situation. We therefore conclude that the observed increase in breeder aggression is caused by our experimental reduction of subordinate digging behaviour and not by conflict over rank or other unintended side effects of the manipulations.

Previous studies of the negotiation between dominant and subordinate group members over the demanded effort of helpers have not differentiated between the various helping behaviours. This is an important shortcoming because these behaviours may involve highly divergent costs for subordinates and benefits for breeders. For example, digging behaviour in N. pulcher was shown to increase routine metabolic rate more than sixfold, whereas aggression is energetically cheaper but might involve injury risk [34,56]. Breeders may benefit from help in defence behaviour mainly when having eggs, larvae or fry [66,67,79], whereas they benefit from saving expensive digging effort throughout the reproductive cycle [34]. Since in our experiment neither eggs, larvae nor fry were present, it is likely that the breeders' demand for help was focused primarily on digging behaviour rather than defence (cf. [66]). Our results are consistent with this hypothesis, as breeders reacted with punishment to reduced digging, but not to reduced defence by helpers. However, it is not entirely clear whether defence by subordinates is part of the negotiation process underlying the pay-to-stay mechanism, even if 'pre-emptive appeasement' by the observed compensatory response of helpers implies this possibility [45]. In any case, our data clearly show that territory maintenance and defence behaviours by helpers are regulated in different ways.

This insight may allow disentangling the effects of previous studies manipulating helping behaviour by confinement. In the field, N. pulcher helpers in small groups were found to receive more aggression from breeders and to show more submission after prevention of helping, whereas their defence efforts increased after release [52]. Our results suggest that in this field study the increased aggression of

breeders might have been caused by the lack of digging support from helpers, and the increased defence of helpers after release may have reflected a compensatory response of helpers to previous idleness, as found in our study. Our results also show that defence-compensation is not restricted to conspecific intruders but also applies to egg predators (cf. [45]). This is remarkable because of the different fitness consequences associated with these divergent types of intruders.

In the competitive social environment characterizing *N. pulcher* colonies, being accepted in a group and maintaining peaceful relationships with social partners is a vital skill [39,57,80–82]. Aggression from group members, apart from the risk of injury and energetic costs, may lead to eviction from the group [40,83]. Such fate is detrimental for subordinates, who cannot survive without the protection of shelters and vigilant group members [39,57]. Stable territories and dominance hierarchies are evidence for conflict reduction through ritualized dominance interactions with aggressive and submissive displays [84]. Our results imply that subordinates can reduce dominant aggression through digging. This territory maintenance behaviour thus seems to have a social function reminiscent of submissive displays. Interestingly, during the manipulation of digging behaviour, subordinates showed less submissive behaviours when digging was prevented than when digging was allowed. As this reduction was achieved by blocking the shelter entrance for the subordinate, it simultaneously reduced its use of this resource. Thus, the reduction of submissive displays in this situation is in line with the previous observation that submissive behaviour is functionally involved in regulating shelter access [85].

Submissive displays are considered to have evolved from flight behaviours or postures of helplessness, through exaggeration and ritualization [84]. Their function is to signal to aggressors that they are not challenged any more and would not gain from further attacks. Our data suggest that digging behaviour has a similar accessory function, as it inhibits aggression from dominant group members. This context-specific addition of function from mere territory maintenance to appeasement prompts questions regarding the involved proximate mechanisms. A look at motion patterns associated with digging and submissive displays reveals a striking similarity. Digging can take two forms, mouth digging (picking up sand with the mouth, carrying it away from the shelter, and spitting it out), and fan digging (propelling sand out of the shelter with the tail fin). The most pronounced and frequent submissive display is tail quivering, where the subordinate swims below the dominant, presses its body against the ground and agitates the tail fin. Tail quivering and fan digging are thus realized by remarkably similar body movements. Interestingly, we frequently observed that a subordinate showing a tail-quivering display switched to fan digging, and in the process even opened its mouth and took up sand, suggesting that there might also be a regulatory connection between these behaviours. The majority of digging behaviours we observed, however, were mouth digging. Hence it is clear that the appeasing effect to breeders of the digging effort of helpers is not merely reflecting a submissive display, like tail quivering. In line with this thought, another experimental study found that breeder aggression also varies with the amount of direct brood care of helpers [51]. Brood care includes providing oxygen to the eggs by agitating the body and tail. This fanning behaviour is also similar to tail quivering, and the two behaviours may merge into each other.

Helping behaviour, according to the pay-to-stay hypothesis, is regulated via an inter-individual feedback loop. Helping reduces aggression from breeders, which in turn rewards helping [48]. This feedback reflects a reciprocal exchange of resources: access to the important resources of the territory and group membership are exchanged for alloparental care. Our data directly support the first part of this hypothesis by experimentally demonstrating that helping via digging reduces aggression. The data are also in support of the second part, that aggression increases helping, although this is less conclusive. The increase in aggression was accompanied not by an increase in digging, but by an increase in other behaviours that reduce aggression, submissive displays. The reason could be that helpers had no opportunity to dig under these high aggression levels, because they were not tolerated in the shelter, and were thus forced to appease dominants in a different way. A negative correlation between either increased helping or increased submission of subordinates in response to breeder aggression was also observed when the participation of helpers in territory defence against conspecific challengers had been prevented [45], which corroborates our interpretation.

Our data provide no unequivocal evidence that territory defence, a behaviour previously classified as helping, is regulated socially through a pay-to-stay mechanism. Even though no eggs or small fry were present during our experiments, helpers defended against the presented egg predator and reacted to our prevention of defence behaviour by increasing defence later on, possibly compensating for the lost opportunity to defend. Although this result is consistent with a social regulatory process where helpers increase defence to appease the breeders, it might also be explained by a purely intra-individual regulation mechanism. Subordinates could benefit from defending the territory against egg predators apart from the

possible appeasing effect on dominant breeders. They occasionally participate in reproduction [38,42,43,83] and may inherit the territory later [29,86]. They might, therefore, benefit from deterring egg predators from the area. This is unlikely, however, as a displacement of highly abundant egg predators from the vicinity of the territory seems unfeasible. Alternatively, they may benefit from the presence of fry even if it is not their own due to group augmentation effects [87,88]. To clearly demonstrate that the compensatory effect is indeed shown for altruistic reasons would require an experiment preventing this compensatory response and measuring the reaction of breeders.

The role of negotiation and trading in animals in general, and in cooperative breeding in particular, has gained increasing support from recent studies [11,14,47], which has importantly supplemented previous models based on ecological constraints and kin selection [89]. Cooperatively breeding primates, insects, birds and mammals exchange commodities in market-like situations [17,21,28,90–92], but while it is easy to record these exchanges in daily behavioural routines it is often difficult to assess whether they are subject to effects of kin selection, mutual immediate fitness benefits (as in mutualisms, [5]), coercion or reciprocity [11,91,93,94]. This is in part due to limits in manipulating the relevant factors experimentally (such as relatedness, environmental challenges, and the behaviours involved) and to difficulties in measuring the relevant fitness consequences for the involved partners. In cooperatively breeding cichlids, these manipulations are possible. The estimation of fitness consequences remains a difficult task that requires a thorough understanding of the ecology and the population-genetic and social structure (including the dynamics between groups of a colony) of the animals [82]. Since many of these parameters are well known in *N. pulcher*, social cichlids provide a unique opportunity to study the relative importance of these factors in detail, including the study of individual helping behaviours [38].

Ethics. Experiments were approved by the Veterinary Office of the Kanton Bern (licence no. 74/15).

Data accessibility. The datasets supporting this article and the code we used for producing the figures and results are available as electronic supplementary material at the journal's online platform.

Authors' contributions. J.N. and M.T. conceived and designed the study. J.N. carried out the experiments, collected the data and carried out the statistical analyses. J.N. wrote the first draft of the manuscript, which was revised by M.T. Both authors gave final approval for publication and agree to be held accountable for the work performed therein.

Competing interests. We declare we have no competing interests.

Funding. The authors acknowledge support of the Swiss National Science Foundation.

Acknowledgements. We thank Michael Cant and Beat Naef-Daenzer for discussion and comments on the manuscript, Evi Zwygart for taking care of the animals and Markus Wymann for helping in the construction of experimental partitions.

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
