## [Reviewer comments · Royal Society Open Science]

Review History

RSOS-191808.R0 (Original submission)

Review form: Reviewer 1

Is the manuscript scientifically sound in its present form?

No

Are the interpretations and conclusions justified by the results?

No

Is the language acceptable?

Yes

Do you have any ethical concerns with this paper?

No

Have you any concerns about statistical analyses in this paper?

Yes

Recommendation?

Major revision is needed (please make suggestions in comments)

Comments to the Author(s)

I maintain that the authors have not analyzed their data appropriately, and that these response variables are indeed not independent. For instance, aggression in one actor shapes submission responses in another, and social interactions are causally linked with both of these other behaviors. These behaviors are causally linked together in a network. Thus, while I appreciate that the authors would like to dissect their analysis behavior-by-behavior, I do not think this is an acceptable method on its own. A combined multivariate test should be conducted first. Following a significant multivariate test, I agree the authors should proceed with a behavior by behavior analysis. Otherwise, they will need to use a modified p-value, which would eliminate some of their more interesting results. I remain firm on this point and do not accept the author's absent argument for rejecting the interdependence of these behaviors.

That aside, it is clear from my reading that there will continue to be interesting results from this paper following a multivariate or network analysis, and that the experiments here are cleverly designed, sound, and deserving of positive attention. I like this paper and these authors are superb writers. They have respected and honored the other major concerns of the other reviewer and myself, and I would like to afford them every opportunity to revise their work.

Review form: Reviewer 2

Is the manuscript scientifically sound in its present form?

Yes

Are the interpretations and conclusions justified by the results?

Yes

Is the language acceptable?

Yes

Do you have any ethical concerns with this paper?

No

Have you any concerns about statistical analyses in this paper?

No

Recommendation?

Accept with minor revision (please list in comments)

Comments to the Author(s)

The authors have done a good job of addressing previous comments and clarifying many of the confusions in the first draft of the manuscript. I think the current draft is very clear and well-written. I have only a few minor comments:

Lines 53-59 and 82-84: I think an example in both these sections would clarify these points.

Line 151: Italics are missing from the species name.

Lines 359-371: I think the authors should explicitly address the fact that there were no fry present in this study here. I know the authors address this in a paragraph earlier in the discussion, but I do not believe it fully addresses the arguments raised in this paragraph, even if their current explanation may still be plausible due to the potential for future reproduction.

Figure 2: What type of post-hoc test was used to calculate differences among treatment groups and how did it control for multiple comparisons? I see the authors' response to Reviewer 2's comment about multiple comparisons, but I think this is a separate issue.

Decision letter (RSOS-191808.R0)

11-Nov-2019

Dear Mr Naef,

The editors assigned to your paper ("Commodity-specific punishment for experimentally induced defection in cooperatively breeding fish") have now received comments from reviewers. We would like you to revise your paper in accordance with the referee and Associate Editor suggestions which can be found below (not including confidential reports to the Editor). Please note this decision does not guarantee eventual acceptance.

Please submit a copy of your revised paper before 04-Dec-2019. Please note that the revision deadline will expire at 00.00am on this date. If we do not hear from you within this time then it will be assumed that the paper has been withdrawn. In exceptional circumstances, extensions may be possible if agreed with the Editorial Office in advance. We do not allow multiple rounds of revision so we urge you to make every effort to fully address all of the comments at this stage. If deemed necessary by the Editors, your manuscript will be sent back to one or more of the original reviewers for assessment. If the original reviewers are not available, we may invite new reviewers.

- Data accessibility

It is a condition of publication that all supporting data are made available either as supplementary information or preferably in a suitable permanent repository. The data accessibility section should state where the article's supporting data can be accessed. This section should also include details, where possible of where to access other relevant research materials such as statistical tools, protocols, software etc can be accessed. If the data have been deposited in

an external repository this section should list the database, accession number and link to the DOI for all data from the article that have been made publicly available. Data sets that have been deposited in an external repository and have a DOI should also be appropriately cited in the manuscript and included in the reference list.

<http://datadryad.org/submit?journalID=RSOS&manu=RSOS-191808>

- **Competing interests**

- **Authors' contributions**

- **Acknowledgements**

- **Funding statement**

Kind regards,

on behalf of Dr Kristina Sefc (Associate Editor) and Kevin Padian (Subject Editor)
openscience@royalsociety.org

Associate Editor's comments (Dr Kristina Sefc):

Dear authors,

Thank you for transferring your manuscript to RSOS. Your revision has been seen by the original reviewers. Reviewer 1 (previously ref. 2) maintains the original criticism of the statistical analysis, and this concern is also picked up by reviewer 2 in their remarks to the editor. Both, however, recommend the manuscript to RSOS. Please follow the suggestions of the referees, and I look forward to receiving your resubmission.

Kind regards,
Kristina Sefc

Reviewers' Comments to Author:

Reviewer: 1
Comments to the Author(s)

I maintain that the authors have not analyzed their data appropriately, and that these response variables are indeed not independent. For instance, aggression in one actor shapes submission responses in another, and social interactions are causally linked with both of these other behaviors. These behaviors are causally linked together in a network. Thus, while I appreciate that the authors would like to dissect their analysis behavior-by-behavior, I do not think this is an acceptable method on its own. A combined multivariate test should be conducted first. Following a significant multivariate test, I agree the authors should proceed with a behavior by behavior analysis. Otherwise, they will need to use a modified p-value, which would eliminate some of their more interesting results. I remain firm on this point and do not accept the author's absent argument for rejecting the interdependence of these behaviors.

That aside, it is clear from my reading that there will continue to be interesting results from this paper following a multivariate or network analysis, and that the experiments here are cleverly designed, sound, and deserving of positive attention. I like this paper and these authors are superb writers. They have respected and honored the other major concerns of the other reviewer and myself, and I would like to afford them every opportunity to revise their work.

Reviewer: 2
Comments to the Author(s)

The authors have done a good job of addressing previous comments and clarifying many of the confusions in the first draft of the manuscript. I think the current draft is very clear and well-written. I have only a few minor comments:

Lines 53-59 and 82-84: I think an example in both these sections would clarify these points.

Line 151: Italics are missing from the species name.

Lines 359-371: I think the authors should explicitly address the fact that there were no fry present in this study here. I know the authors address this in a paragraph earlier in the discussion, but I do not believe it fully addresses the arguments raised in this paragraph, even if their current explanation may still be plausible due to the potential for future reproduction.

Figure 2: What type of post-hoc test was used to calculate differences among treatment groups and how did it control for multiple comparisons? I see the authors' response to Reviewer 2's comment about multiple comparisons, but I think this is a separate issue.

Author's Response to Decision Letter for (RSOS-191808.R0)

See Appendix A.

RSOS-191808.R1 (Revision)

Review form: Reviewer 1

Is the manuscript scientifically sound in its present form?

Yes

Are the interpretations and conclusions justified by the results?

Yes

Is the language acceptable?

Yes

Do you have any ethical concerns with this paper?

No

Have you any concerns about statistical analyses in this paper?

No

Recommendation?

Accept as is

Comments to the Author(s)

First, I must apologize to the editor and authors for my tardy review. I'm in the midst of a long-term field season and I'm only now circling back to honoring my commitment to this review. I am, however, happy to convey that I am pleased with the authors' revised analyses and interpretation. This paper is certainly deserving of publication in Roy Soc Open Science, and I am now happy to give it my most enthusiastic endorsement. I thank the authors for taking the extra effort to address my nagging concerns. They've produced a clever study they can be very proud of.

Decision letter (RSOS-191808.R1)

14-Jan-2020

Dear Mr Naef,

It is a pleasure to accept your manuscript entitled "Commodity-specific punishment for experimentally induced defection in cooperatively breeding fish" in its current form for publication in Royal Society Open Science. The comments of the reviewer(s) who reviewed your manuscript are included at the foot of this letter.

on behalf of Dr Kristina Sefc (Associate Editor) and Kevin Padian (Subject Editor)
openscience@royalsociety.org

Reviewer comments to Author:
Reviewer: 1

Comments to the Author(s)

First, I must apologize to the editor and authors for my tardy review. I'm in the midst of a long-term field season and I'm only now circling back to honoring my commitment to this review. I am, however, happy to convey that I am pleased with the authors' revised analyses and interpretation. This paper is certainly deserving of publication in Roy Soc Open Science, and I am now happy to give it my most enthusiastic endorsement. I thank the authors for taking the extra effort to address my nagging concerns. They've produced a clever study they can be very proud of.

Appendix A

Response to Referees

Replies to Referee 1

I maintain that the authors have not analyzed their data appropriately, and that these response variables are indeed not independent. For instance, aggression in one actor shapes submission responses in another, and social interactions are causally linked with both of these other behaviors. These behaviors are causally linked together in a network. Thus, while I appreciate that the authors would like to dissect their analysis behavior-by-behavior, I do not think this is an acceptable method on its own. A combined multivariate test should be conducted first. Following a significant multivariate test, I agree the authors should proceed with a behavior by behavior analysis. Otherwise, they will need to use a modified p-value, which would eliminate some of their more interesting results. I remain firm on this point and do not accept the author's absent argument for rejecting the interdependence of these behaviors.

Reply: We have now conducted the non-parametric multivariate tests the reviewer suggested in the first reviewing round (PERMANOVA from the R-package vegan). The results confirm our conclusions, and the respective information was added to the methods and results sections (lines 206-218; 222-226; 252-255). Where the multivariate test did not indicate significant treatment effects, the previous analyses of individual behaviours were removed from the text and the results from the multivariate test was provided instead. This was the case for the “no-demand” control situation (lines 252-255) and for the effect of defence prevention during the manipulation phase (lines 225-226, 236-239 and 247-249).

That aside, it is clear from my reading that there will continue to be interesting results from this paper following a multivariate or network analysis, and that the experiments here are cleverly designed, sound, and deserving of positive attention. I like this paper and these authors are superb writers. They have respected and honored the other major concerns of the other reviewer and myself, and I would like to afford them every opportunity to revise their work.

Reply: Thank you very much for these encouraging comments.

Replies to Referee 2

The authors have done a good job of addressing previous comments and clarifying many of the confusions in the first draft of the manuscript. I think the current draft is very clear and well-written. I have only a few minor comments:

Reply: We are grateful for this encouraging evaluation and thank the reviewer for the constructive comments.

Lines 53-59 and 82-84: I think an example in both these sections would clarify these points.

Reply: We agree and have added an example in each of these two paragraphs (lines 52-53 and 81-83). Further examples can be found in the references provided in those paragraphs.

Line 151: Italics are missing from the species name.

Reply: This was corrected (line 150).

Lines 359-371: I think the authors should explicitly address the fact that there were no fry present in this study here. I know the authors address this in a paragraph earlier in the discussion, but I do not believe it fully addresses the arguments raised in this paragraph, even if their current explanation may still be plausible due to the potential for future reproduction.

Reply: We agree that this paragraph benefits from explicitly mentioning that there were no fry present. We have now added this (lines 362-365).

Figure 2: What type of post-hoc test was used to calculate differences among treatment groups and how did it control for multiple comparisons? I see the authors' response to Reviewer 2's comment about multiple comparisons, but I think this is a separate issue.

Reply: The statistical procedure was now adjusted in accordance with the suggestions of reviewer 1 (previously reviewer 2). We now employ a multivariate test to assess how our treatments affected the behavioural composition, prior to the analyses of treatment effects of individual behaviours (lines 212-216). We only continued with the analysis of individual behaviours where the multivariate test indicated significant treatment effects. For the analyses of individual behaviours, we used generalized linear mixed models for negative binomial data. In this two-step procedure, we deem a correction for multiple testing too conservative, as the analyses of individual behaviours were based on a priori hypotheses and were conducted only after a significant multivariate test.